# Advancing Cholangiocarcinoma Care: Insights and Innovations in T Cell Therapy

**DOI:** 10.3390/cancers16183232

**Published:** 2024-09-23

**Authors:** Neda Dadgar, Arun K. Arunachalam, Hanna Hong, Yee Peng Phoon, Jorge E. Arpi-Palacios, Melis Uysal, Chase J. Wehrle, Federico Aucejo, Wen Wee Ma, Jan Joseph Melenhorst

**Affiliations:** 1Cleveland Clinic Foundation, Enterprise Cancer Institute, Translational Hematology & Oncology Research, Cleveland, OH 44114, USA; dadgarn@ccf.org; 2Cleveland Clinic Foundation, Lerner Research Institute, Center for Immunotherapy and Precision Immuno-Oncology, Cleveland, OH 44195, USA; arunaca@ccf.org (A.K.A.); hongh2@ccf.org (H.H.); phoony@ccf.org (Y.P.P.); arpij3@ccf.org (J.E.A.-P.); uysalm@ccf.org (M.U.); 3Cleveland Clinic Foundation, Digestive Diseases & Surgery Institute, Cleveland, OH 44195, USA; wehrlec@ccf.org (C.J.W.); aucejof@ccf.org (F.A.); 4Cleveland Clinic Foundation, Taussig Cancer Institute, Cleveland, OH 44106, USA; maw4@ccf.org

**Keywords:** CCA, CAR T cell therapy, immunotherapy, tumor-infiltrating lymphocytes, precision medicine

## Abstract

**Simple Summary:**

This review article explores the potential of CAR T cell therapy in treating cholangiocarcinoma (CCA), a highly aggressive and difficult-to-treat cancer of the bile ducts. With conventional therapies offering limited success, this article highlights the innovative approach of using immune cell therapy, particularly CAR T cells, to improve outcomes for CCA patients. The review discusses the tumor microenvironment’s role in disease progression and the challenges of implementing CAR T cell therapy in solid tumors. By summarizing current research, clinical trials, and the evolving landscape of precision medicine, this article aims to provide a comprehensive understanding of the future prospects for immunotherapy in CCA, emphasizing the need for continued advancements in this promising field.

**Abstract:**

Cholangiocarcinoma (CCA) is a rare and aggressive malignancy originating from the bile ducts, with poor prognosis and limited treatment options. Traditional therapies, such as surgery, chemotherapy, and radiation, have shown limited efficacy, especially in advanced cases. Recent advancements in immunotherapy, particularly T cell-based therapies like chimeric antigen receptor T (CAR T) cells, tumor-infiltrating lymphocytes (TILs), and T cell receptor (TCR)-based therapies, have opened new avenues for improving outcomes in CCA. This review provides a comprehensive overview of the current state of T cell therapies for CCA, focusing on CAR T cell therapy. It highlights key challenges, including the complex tumor microenvironment and immune evasion mechanisms, and the progress made in preclinical and clinical trials. The review also discusses ongoing clinical trials targeting specific CCA antigens, such as MUC1, EGFR, and CD133, and the evolving role of precision immunotherapy in enhancing treatment outcomes. Despite significant progress, further research is needed to optimize these therapies for solid tumors like CCA. By summarizing the most recent clinical results and future directions, this review underscores the promising potential of T cell therapies in revolutionizing CCA treatment.

## 1. Introduction

Cholangiocarcinoma (CCA) is a highly malignant neoplasm that arises from the biliary epithelium and is characterized by its late presentation and aggressive course [1,2]. The highest incidence rates are reported in Southeast Asia (0.1–1.8/100,000), which are comparatively higher than the 0.6–1/100,000 incidence reported in the United States. CCA is a heterogeneous entity, and can present as an intrahepatic, a perihilar, or a distal extrahepatic tumor [2], each with its own distinct clinical characteristics, prognosis, and management challenges. These variants necessitate a tailored understanding and approach to treatment. Due to the lack of specific and early onset symptoms, nearly 75% of the patients present with locally advanced or metastatic disease and, thus, have limited treatment options [2]. For patients who undergo resection, reported 5-year survival rates are low, ranging from 21 to 35% [3]. These percentages emphasize the urgent need for advanced diagnostic strategies and more effective therapeutic interventions. Moreover, the complex interplay of risk factors, such as chronic inflammation, biliary diseases, and genetic predisposition, highlights the importance of resolving the disease’s etiology and identifying opportunities for prevention and early intervention.

Traditionally, therapeutic options for CCA have been limited, including surgical resection, when diagnosed early, plus systemic chemotherapy. However, recent years have witnessed remarkable progress in the field of CCA therapy [1]. These advancements, including the development of targeted therapies and novel immunotherapeutic approaches, have begun to show potential in improving survival rates and quality of life for patients. For instance, targeted therapies that inhibit specific molecular pathways altered in CCA—such as FGFR2, KRAS, PTEN, CDKN2B, ERBB3, MET, NRAS, CDK6, BRCA1, BRCA2, NF1, PIK3CA, PTCH1, and TSC1—have shown promise in clinical trials, offering hope for more effective management of this challenging disease [4,5]. With the advancement in sequencing techniques and improved understanding of the various molecular pathways in CCA, both targeted therapy and immunotherapy have emerged as promising avenues [6]. Immune cell therapy in the form of immune checkpoint inhibitors (ICIs) [7], cancer vaccines [8], chimeric antigen receptor T cells (CAR T cells) [9], and tumor-infiltrating leukocytes (TILs) [10] are the major immunotherapy methods currently evaluated in CCA. This review explores the role of immune cell therapy in the management of CCA, focusing specifically on CAR T cells and TILs. A thorough understanding of the immune microenvironment within CCA is critical as it offers insights into how T cell therapy can enhance the immune response against cancer cells, potentially shifting treatment outcomes towards more effective management. Real-world successes of T cell therapy are showcased through clinical trials and compelling case studies. Despite recent advancements, T cell therapy is not without its challenges and complexities, which must be navigated to optimize its use in CCA treatment. Hence, this review elucidates the concepts and potential role of immune cell therapy in CCA, summarizes our understanding from various clinical trials, and addresses some of the potential challenges associated with the current immunotherapy options.

## 2. Cholangiocarcinoma: Understanding the Challenge

The highly aggressive nature of CCA is associated with a multitude of risk factors, along with complex and dynamic interactions between malignant cells, stroma, and immune cells in the tumor microenvironment (TME) [6]. The vast inter- and intra-tumoral heterogeneity further complicates patient care management in CCA. Beyond morphological variations and growth patterns, outcomes are also influenced by nucleic acid level changes, including patient-specific driver mutations, gene expression and methylation patterns, and tumor–immune cell interactions within the TME [11]. These factors are critical as they directly impact the effectiveness of therapeutic interventions and the overall prognosis of the disease.

CCA has numerous subtypes with different origins. Intrahepatic CCA (iCCA) originates within the liver parenchyma; perihilar CCA (pCCA) occurs at the confluence of the left and right hepatic ducts; and distal CCA (dCCA) develops in the lower portion of the bile duct near the duodenum. CCA can also be classified into three main patterns of growth based on its gross appearance: mass-forming (MF), periductal infiltrating (PI), and intraductal growing (IG) [12]. The MF type presents as a mass lesion in the hepatic parenchyma. The PI type grows longitudinally along the wall of large bile ducts, and the IG type presents as a polypoid or papillary tumor growing towards duct lumina [12]. Understanding these distinct patterns is crucial for the diagnosis and tailoring personalized treatment approaches to the specific needs of individual patients [13]. For instance, PI patterns may require more aggressive surgical intervention due to their tendency to spread along the bile duct and high rates of lymph node metastasis, whereas IG tumors, which are often more localized, might be candidates for less extensive surgical procedures or localized therapies [13]. Recognizing these patterns not only assists in choosing the appropriate therapeutic strategy but also enhances the prognosis by aligning treatments with the tumor’s behavior and the patient’s clinical profile. The success of targeted therapies underlines the importance of identifying the underlying genetic variations using comprehensive sequencing technologies. However, obtaining tumor tissue samples in the early stages of CCA remains challenging, particularly due to the presence of desmoplastic stroma and difficulty in accessing bile ducts for brush cytology. This dense, fibrous tissue forms around the tumor, significantly complicating biopsy procedures by limiting access to the malignant cells. Early sampling is essential as it allows for better characterization of the tumor in CCA, leading to more precisely targeted therapies and potentially improving patient outcomes.

The etiology of CCA is multifactorial, with a range of risk factors contributing to its development. Chronic inflammation of the bile ducts, often associated with conditions such as primary sclerosing cholangitis (PSC) and chronic biliary infections, is a well-established risk factor [14]. Additionally, exposure to toxins, such as certain chemicals and liver flukes, genetic predisposition, and underlying liver diseases, including cirrhosis, can heighten the risk of CCA [15]. One of the greatest challenges in managing CCA lies in its insidious nature, with symptoms often remaining undetectable until the disease has advanced to later stages. Common clinical presentations include jaundice, abdominal pain, unexplained weight loss, and changes in stool or urine color [16]. Recognizing these signs and symptoms early on is pivotal for timely diagnosis and intervention.

The effectiveness of immune cell therapy in CCA largely depends on the fitness and distribution of the immune cells within the TME. These factors are crucial for determining which patients are likely to benefit from such treatments. However, the ability to identify predictive biomarkers that can accurately gauge immune cell readiness and positioning within the TME is currently limited. This limitation poses a significant challenge in selecting suitable patients for therapy and highlights the need for continued research into more reliable biomarkers. Given these complexities, this review aims to explore current treatment options, address the challenges in managing CCA, and underscore the importance of advancing our understanding and methodologies in this area.

## 3. Cholangiocarcinoma and the Immune Microenvironment

CCA, especially the intrahepatic variant, is characterized by a prominent TME, marked by abundant fibroblast-induced desmoplastic reactions [17]. The TME of CCA consists of a diverse range of cells, including stromal cells like cancer-associated fibroblasts (CAFs), endothelial cells, and immune cells from both the innate and adaptive immune systems such as tumor-associated macrophages (TAMs), neutrophils, natural killer cells, and T and B lymphocytes (Figure 1) [18,19,20]. Apart from the tumor, immune, and stromal cells, other components in the TME, like extracellular vesicles, soluble molecules, and cytokines secreted from the cells, also play a critical role in modulating the behavior of cancer cells. These components contribute to cancer progression by facilitating communication between cells, promoting tumor growth, and suppressing the immune response. Additionally, they influence the response to treatments by affecting drug delivery and the efficacy of immunotherapies [21]. The TME of CCA, which is highly enriched with immunosuppressive populations like myeloid and monocyte-derived suppressor cells (MDSCs), TAMs, tumor-associated neutrophils (TANs), and regulatory T (Treg) cells, plays an important role in its aggressive nature [22]. The dynamic interaction between different cell types in the immunosuppressive environment forms a potential barrier to the immune cells and immunotherapy, leading to poor treatment outcomes [23]. While the old school of thought has proposed targeting the tumor stroma as a potential strategy to combat CCA, recent findings in pancreatic cancer have raised questions about this approach [24]. These findings emphasize the importance of understanding the complex interactions within the TME, which influence both tumor progression and the efficacy of treatments. Appendix A illustrates the clinical relevance of different immune cells in the TME, summarizing findings from CCA patients. Given the intricate roles played by both the innate and adaptive immune systems in the TME, a thorough review of these systems is crucial. This review aims to elucidate how both the innate and adaptive immune cells contribute to the immune landscape of CCA, affecting tumor growth, suppression, and response to therapies.

### 3.1. Innate Immune System

The innate immune system plays a crucial role in CCA progression. TAMs in CCA are primarily of the M2 phenotype, influenced by the STAT3 pathway, and release cytokines (TNF-α, TGF-β, IL-6, IL-10, VEGF-A) that support tumor growth and metastasis [25]. Recruitment of innate immune cells involves various chemoattractant molecules and cytokines, like MCP-1/CCL2, CSF-1, and VEGF-A [26]. CCA cells induce the polarization of macrophages towards the tumor-promoting M2 phenotype, leading to the release of inflammatory and growth factors that support tumor growth. TAMs and TANs contribute to tumor progression by promoting angiogenesis, releasing pro-inflammatory mediators, and suppressing anti-tumor immune responses [27]. For example, tumor-derived factors such as GM-CSF and chemokines induce proliferation, expansion, and recruitment of both monocytic (M-MDSCs) and myeloid (PMN-MDSCs) suppressor cells [28]. These MDSCs have potent immunosuppressive activity on both innate and adaptive immunity [29,30], leading to resistance and treatment failure. Basophils usually are a very small subset in the TME, and limited data are available on their clinical impact, with animal models showing both pro- and anti-tumorigenic properties [31].

### 3.2. Adaptive Immune System

Unlike the innate immune system, the adaptive immune system plays a significant role in recognizing and targeting emerging tumor cells, acting as a primary defense against cancer. Over the last decade, cell-based immunotherapy, such as TILs, has gained significant attention in terms of providing an alternate strategy to treat solid tumors. TILs are immune cells found in many solid tumors consisting of various cell types, including B lymphocytes, CD8+ cytotoxic T lymphocytes, CD4+ T helper lymphocytes, and Tregs [22]. In CCA, CD8+ T lymphocytes are predominantly located within the tumor tissue, while CD4+ TILs are more prevalent in the surrounding peri-tumoral region [32]. Studies have shown that an increased presence of CD4+ and CD8+ TILs in CCA is associated with better overall survival, fewer lymph node metastases, and reduced invasion [33,34,35]. Conversely, low numbers of CD8+ TILs are linked to worse overall survival [36]. Additionally, the expression of MHC class I molecules in CCA correlates strongly with the presence of CD4+ and CD8+ TILs and is associated with prolonged survival [33]. Dendritic cells (DCs) are found both within the tumor core and at the invasive front, with mature DCs mainly located at the tumor edge [37]. They play a crucial role by bridging the adaptive and innate immune responses. Mature DCs interact with CD4+ and CD8+ T cells at the cancer periphery, emphasizing the importance of immune cell interactions in CCA [37]. Patients with higher numbers of mature DCs at the tumor–host interface tend to have better prognoses and lower rates of lymph node metastases. On the other hand, an increase in plasmacytoid dendritic cells (pDCs) has been associated with poor survival in solid tumors, although limited data are available specific to CCA [38,39]. The role of B lymphocytes in CCA remains unclear, as they are rarely observed in patient tissues. While high densities of CD20+ B cells in low-grade tumors are associated with favorable overall survival, further research is needed to fully understand their significance in CCA [40].

CCA tumor cells exhibit multiple immune evasion mechanisms that contribute to the aggressive nature of the disease and poor outcomes. These mechanisms include the secretion of immunosuppressive signals like TGF-β, CCL2, and FoxP3, which promote the transformation of tumor-infiltrating T cells into Tregs that secrete immunosuppressive cytokines, creating an immunosuppressive microenvironment [18]. TGF-β, which is overexpressed in CCA, plays a dual role, influencing both tumor growth and immune regulation. FoxP3 overexpression in CCA cells and CTLA-4 expression on Tregs contribute to immune escape. CCA cells express ligands (e.g., PD-L1) and receptors that interact with immune checkpoints like PD-1 and CTLA-4 on the T cells, correlating with poor prognosis.

These mechanisms collectively hinder effective immune surveillance, emphasizing the need for innovative therapeutic strategies to counteract immune evasion in CCA. Recent research has illuminated the pivotal role of pro- and anti-tumorigenic immune cells in the TME that modulate and regulate CCA development and progression. Simultaneously, studies show the pro-invasive functions of stromal cells, particularly CAFs, supporting not only cancer growth and invasion but also early dissemination through lymphatic vessels [41,42]. Given these complexities of CCA, effective and personalized therapies should encompass multimodal strategies, targeting of stromal cells, and leveraging high-throughput screening and patient stratification based on next-generation sequencing for innovative therapeutic approaches.

## 4. Advancements in Immunotherapy and Cellular Dynamics: Targeting Cholangiocarcinoma for Improved Therapeutic Strategies

Recent advancements in the understanding of CCA have been significantly driven by immunohistochemical (IHC) techniques and molecular tissue imaging. These methodologies have been crucial in explaining the complex molecular pathways, immune landscapes, and, particularly, the role and therapeutic potential of T cells within different subtypes of CCA. Appendix A summarizes the IHC markers pertinent to the exploration of T cell therapy in CCA.

Key studies leveraging IHC techniques and molecular imaging have highlighted the significance of T cell markers and responses in the context of CCA. For instance, Chung’s work, although initially focused on the PI3K/AKT pathway, also showed the importance of evaluating T cell infiltration and activation status as a basis for targeted T cell therapies [43]. Similarly, Huang’s research into the immune microenvironment of Epstein–Barr virus-associated intrahepatic cholangiocarcinoma (EBVaICC) revealed the diagnostic and therapeutic relevance of T cell markers, which is crucial for developing targeted T cell therapies. Understanding these dynamics in EBVaICC, where the Epstein–Barr virus influences cancer development and interacts with the immune system, provides vital information for designing effective immunotherapies [44]. Sturm’s exploration of biliary strictures with imaging strategies points towards the differentiation of ICC using specific markers, which could be extended to identify T cell subpopulations and their roles in the tumor microenvironment [45]. Tian’s examination of PD-1/PD-L1 expression profiles within ICC underscores the potential of targeting these immune checkpoints to enhance T cell responses [46].

Moreover, Kim’s investigation into the prognostic implications of tumor-infiltrating FoxP3- CD4+ T cells in biliary tract cancer illustrates the critical role of these T cells in CCA. Unlike FoxP3+ CD4+ T cells, which typically exhibit regulatory functions that can suppress anti-tumor immune responses, FoxP3- CD4+ T cells can enhance anti-tumor activity. Understanding the balance and function of these T cell subsets in the T cell microenvironment is crucial for developing therapies that effectively modulate this balance, enhancing the body’s immune response against cancer [47]. Xia’s comprehensive study mapping the immune cell atlas across CCA subtypes provides a detailed overview of T cell populations that could guide the development of T cell-targeted interventions [48]. Appendix A provides a summary of the research studies that employed immunohistochemistry (IHC) techniques.

## 5. The Promise of T Cell Therapy in Cancer Treatment

The success of T cell therapy in CCA hinges on T cells effectively recognizing and responding to tumor antigens to attack cancer cells. While traditional treatments like surgery, chemotherapy, and radiation have limited benefits in certain patient populations, T cell therapy represents a potential complementary approach that leverages the immune system to improve outcomes. The progression of cancer correlates with tumor cells escaping immunological surveillance via downregulation of MHC class I, antigen loss, and microenvironment changes, such as increased Tregs, MDSCs, and M2 macrophages. Immune checkpoint blockade [49], therapeutic vaccination [50], adoptive transfer of genetically modified CD4+ T cells [51], and CAR T cell therapy [52] are some approaches that utilize T cells for cancer immunotherapy, as outlined in Table 1. These strategies aim to harness the full potential of both CD4+ and CD8+ T cells to enhance anti-tumor responses. While initial studies focused on cytotoxic CD8 T cells, now CD4+ T cells are known to play a significant role in cancer immunotherapy, and various strategies are being explored to leverage their potential in treating different cancer types. CD8+ T cells are known for their direct tumor cell recognition and destruction, while CD4+ T cells support and sustain CD8+ T cell activity. In addition, CD4+ T cells also contribute to the activation of Antigen-Presenting Cells (APCs) and promote the expansion of memory T cells [51]. Notably, CD4+ T cells have been found to recognize mutant neoantigens in cancers, and their responses can lead to anti-tumor effects [53]. Additionally, studies have demonstrated that CD4+ T cells can significantly enhance anti-tumor immunity in other cancer types. In colorectal cancer, CD4+ T cells contribute to a more robust immune response within the tumor microenvironment, aiding in cancer control and potential regression [54]. Similarly, in melanoma, modifications in CD4+ T cell functions, such as the fucosylation of HLA-DRB1, have been shown to improve the efficacy of immunotherapies, leading to better patient outcomes [54].

CD4+ T cells play a dual role in tumor elimination: they not only support CD8+ T cells but also can directly participate in destroying tumor cells. CD8+ T cells, known as cytotoxic effector cells, can become exhausted after prolonged exposure to tumor antigens, which diminishes their ability to effectively kill tumor cells [51]. This exhaustion highlights the importance of CD4+ T cells, as they are crucial for reactivating and maintaining the cytotoxic activity of CD8+ T cells through the release of cytokines like IFN-γ. When CD4+ T cells encounter MHC class II molecules, they release IFN-γ, which has multiple anti-tumor effects. First, it stimulates the production of chemokines Mig and IP-10, which are anti-angiogenic and contribute to the destruction of tumor vasculature, thereby inhibiting tumor growth and inducing necrosis. Furthermore, IFN-γ activates macrophages to release TNF and nitric oxide, enhancing the inflammatory response against the tumor [55] (Table 1).

The expansion in adoptive T cell therapy (ACT) and CAR T cell therapy for CCA reflects the growing understanding of the critical roles played by T cells, especially in their selection, isolation, and expansion [56]. CAR T cell therapy involves genetically modifying the T cells to recognize and target specific tumor antigens in an MHC-independent manner [57]. Other forms of immunotherapy include checkpoint blockade, which involves blocking inhibitory molecules, for example, PD-1 and CTLA-4, to enhance the endogenous T cell activation against tumors, and cancer vaccines, which stimulate T cell responses by introducing tumor-associated antigens to the immune system. However, it is important to note that although T cell therapy offers immense promise in reshaping the landscape of solid cancer treatment, the outcomes have generally been not as impressive as seen in hematological malignancies.


cancers-16-03232-t001_Table 1Table 1Overview of key immunotherapeutic strategies in cancer treatment: exploring the role of T Cells, advanced therapies, and immune modulation techniques. Abbreviations: adoptive cell therapy (ACT), chimeric antigen therapy (CAR T), cytotoxic T lymphocyte-associated protein 4 (CTLA-4), helper T cell (Th), interferon (IFN), programmed cell death protein 1 (PD-1).Key ComponentDescriptionImpact on Cancer TreatmentReferencesRoles of CD4+ and CD8+ T CellsCD8+ T cells directly recognize and destroy tumor cells; CD4+ T cells support CD8+ activity and help in the activation of Antigen-Presenting Cells.Enhances the body’s immune defense against cancer by eliminating abnormal cells and supporting immune response.
ACT and CAR T Cell TherapiesACT involves using auto or allogenic immune cells to treat cancer or infection; CAR T cell therapy involves genetically modifying T cells to recognize tumor antigens.Directly targets and eradicates tumor cells with tumor-specific T cells, offering personalized cancer treatment.[56]Immune Checkpoint BlockadeBlocks inhibitory molecules like PD-1 and CTLA-4 to enhance T cell activation against tumors.Improves T cell activation and ability to combat cancer cells, overcoming cancer’s evasion mechanisms.[58]Therapeutic VaccinationStimulates T cell responses by introducing tumor-associated antigens to the immune system.Aims to prevent cancer development or recurrence by enhancing the immune system’s ability to recognize and destroy cancer cells.[59,60]Dendritic Cells and T Cell ActivationDendritic cells present tumor antigens to T cells, leading to their activation and the targeting of cancer cells.Crucial for initiating a targeted immune response against cancer, leading to the destruction of cancer cells.[60,61,62]Differentiation of CD4+ T Cells into Th SubsetsCD4+ T cells can differentiate into various subsets like Th1, Th2, and Th17, based on cytokine profiles, contributing differently to the immune response.Balances the immune response to cancer, with potential implications for reducing autoimmunity and enhancing tumor targeting.[51]Importance of IFN-γ and ChemokinesIFN-γ released by CD4+ T cells induces chemokines with anti-angiogenic effects, destroying tumor vasculature.Plays a significant role in inhibiting tumor growth and causing necrosis, enhancing the effectiveness of the immune response against cancer.[55]


## 6. Exploring the Role of Immunotherapy in Cholangiocarcinoma

Immunotherapy, a groundbreaking approach in oncology, has introduced a paradigm shift in the treatment of various cancers, including CCA [63]. Notably, checkpoint inhibitors, including PD-1/PD-L1 inhibitors, have garnered significant attention for their ability to expose cancer cells to the immune system, thereby prompting anti-tumor responses [58]. This shift towards immunotherapy marks a typical change in CCA treatment, now considered a standard of care for many patients despite its variable efficacy across individuals (Appendix A). For instance, studies such as the TOPAZ trial and subsequent Italian trials have demonstrated the superiority of combining chemotherapy with immunotherapy over chemotherapy alone in advanced CCA settings [62,64].

Cancer vaccines, which are still in the research phase for bile duct cancer, aim to trigger an immune response against cancer-specific antigens [8,59,60]. Adoptive cell transfer, including the CAR T cell therapy that directly targets cancer cells, offers a promising approach given the complex solid tumor microenvironment of CCA [65,66]. Appendix A provides an overview of ongoing clinical trials investigating various immunotherapeutic approaches for CCA. These trials cover a broad spectrum of strategies, including checkpoint inhibitors targeting PD-1/PD-L1 and CTLA-4, as well as innovative combinations with chemotherapy, targeted therapies, and other immunomodulators. Notable among these are the trials evaluating the efficacy of nivolumab in combination with stereotactic ablative radiation therapy after induction chemotherapy (NCT04648319), durvalumab and tremelimumab combined with platinum-based chemotherapy for intrahepatic CCA (NCT04989218), and pembrolizumab combined with gemcitabine and cisplatin as perioperative therapy for potentially resectable intrahepatic CCA (NCT05967182). Additionally, there are investigations into the use of other checkpoint inhibitors, such as Atezolizumab (NCT05000294) and Sintilimab (NCT05010681), both as monotherapy and in combination with other agents. The diverse range of trials reflects a multifaceted strategy, encompassing checkpoint inhibitors, combination therapies, and novel agents, with the aim of improving outcomes for patients with bile duct cancer.

## 7. Specific Antigen Expression Profiles in CCA

Despite the potential of ICIs, their efficacy in CCA can be limited by the variability in immune response among patients. Targeting the specific antigen expression profiles in CCA presents a significant opportunity in this regard. Table 2 and Figure 2 summarize the antigen expression profiles associated with CCA and their implications for the advancement of precision medicine.

The pursuit of more effective diagnostic and therapeutic options for CCA has led researchers to explore specific biomarkers that are exclusively or predominantly expressed in this cancer type. Among these, carcinoembryonic antigen (CEA) emerges as a notable biomarker, with its glycoprotein overexpression holding promise as a diagnostic indicator [67,68,69,70]. Various antibodies targeting CEA, such as cibisatamab, altumomab pentetate, besilesomab, and labetuzumab, are currently under investigation for their diagnostic and therapeutic potential. Similarly, CA19-9 is recognized for its elevated levels in the blood of CCA patients, serving both as a diagnostic tool and a means for disease monitoring [67,68,71,72,73,74,75,76,77], with ongoing evaluations of monoclonal antibodies targeting this marker in bile duct and colorectal cancers [74]. Furthermore, mucin 1 (MUC1), overexpressed in select CCA cases, is under investigation as a potential therapeutic target, with several antibodies in the pipeline, including CanAg, cantuzumab mertansine, clivatuzumab tetraxetan, gatipotuzumab, pankomab, and pemtumomab [9,78,79,80,81,82]. Additionally, epithelial cell adhesion molecule (EpCAM) is another target for immunotherapy in CCA, with therapies such as catumaxomab exploring its potential [83,84,85]. Epidermal growth factor receptor (EGFR), overexpressed in certain CCA cases, stands out as a potential target for tailored therapies. Investigative avenues include agents like sorafenib, amivantamab, cetuximab, and others [65,81,86,87,88,89,90,91].


cancers-16-03232-t002_Table 2Table 2Table summarizing key antigens associated with intrahepatic CCA (CCA), with potential as therapeutic targets, available targeting therapies for these antigens, and relevant references.AntigenMonoclonal Antibody/Antibody–Drug Conjugates/BITEs/Other Forms of Targeted TherapiesRef.Carcinoembryonic antigen (CEA)CibisatamabAltumomab pentetateBesilesomabLabetuzumab[67,68,69,70]Carbohydrate antigen 19-9 (CA19-9)MVT-5873 (monoclonal antibody targeting CA19.9)[74]Mucin 1 (MUC1)Cantuzumab mertansineClivatuzumab tetraxetanGatipotuzumabPemtumomab[9,78,79,80,81,82]Epithelial cell adhesion molecule (EpCAM)Catumaxomab[83,84,85]Prominin-1 (CD133)AC133-vcMMAF (antibody–drug conjugate)[65,83,92]Epidermal growth factor receptor (EGFR)CetuximabDepatuxizumab mafodotinFutuximabImgatuzumabLaprituximab emtansineMatuzumabNecitumumabPanitumumabZalutumumab[65,81,86,87,88,89,90,91]Glypican-3 (GPC3)Codrituzumab[93,94]Prostate-specific membrane antigen (PSMA)PSMA radioligand therapy[95,96]Wilms tumor 1 (WT1) WT1 vaccines and dendritic cell therapy[60,61,97,98]Human epidermal growth factor 2 (HER2)Trastuzumab Trastuzumab duocarmazine[86,87,88,99]Mucin 4 (MUC4)Small molecule inhibitors[81,100,101]Programmed cell death ligand 1 (PD-L1)Atezolizumab[102]Dickkopf-related protein 1 (DKK1)Small molecule and macromolecule inhibitors[76,103]MesothelinAmatuximab[104,105,106]Glypican-1Antibody–drug conjugates[107]Fibroblast growth factor receptor 2 (FGFR2)Aprutumab ixadotinBemarituzumab[108,109]CD73AB680[110]c-METTivantinib[111]


Other antigens, including HER2, CD133, glypican-3 (GPC3), mesothelin, integrin αvβ6 [112], glypican-1, PSMA, and WT1 are also being investigated for their potential as therapeutic targets or markers for CCA [60,61,65,83,86,87,88,93,94,95,96,97,98,99,113,114,115,116,117,118]. The presence of CD133, associated with cancer stem cells in CCA, suggests its role in tumor progression, and the benefits of targeting this antigen are currently being explored [65,83].

The detailed profiling of antigen expression in CCA benefits CAR T cell therapy, which can be custom-designed to target these specific antigens. Antigens such as carcinoembryonic antigen (CEA), CA19-9, MUC1, EpCAM, and EGFR are overexpressed in CCA, presenting unique targets for CAR T cell engineering. By genetically modifying T cells to express chimeric antigen receptors (CARs) that recognize these specific antigens, it may be possible to direct the potent cytotoxic activity of T cells specifically against CCA cells expressing these markers. For instance, targeting CEA or CA19-9 with CAR T cells could exploit their high expression in CCA to achieve targeted tumor cell eradication while minimizing harm to normal tissues. Similarly, engineering CAR T cells against MUC1, EpCAM, and EGFR could harness their roles in tumor progression and resistance to conventional therapies, offering a novel approach to overcome these challenges. By focusing on these highly specific antigens, CAR T cell therapy has the potential to significantly enhance the therapeutic landscape for CCA in a more effective, targeted, and personalized manner.

## 8. Chimeric Antigen Receptor (CAR) T Cell Therapy for CCA

CAR T cell therapy has emerged as a promising immunotherapeutic approach for cancer treatment, primarily in hematologic malignancies characterized by CD19-positive tumors [119,120]. In contrast, its application to solid tumors has presented significant challenges. However, recent clinical studies in various solid tumors, including glioblastoma [121] and ovarian cancer [122], have shown encouraging results, signaling the potential of CAR T cell therapy in the realm of solid malignancies. The evolution of CAR T technology from first-generation CARs, which had limited efficacy, to second- and third-generation CARs has shown improved clinical responses [123]. Additionally, the advent of fourth-generation armored CARs, known as TRUCKs (T cells redirected for universal cytokine killing), aims to enhance the effectiveness of CAR T therapy in solid tumors by recruiting additional immune cells through the secretion of immunomodulatory cytokines [124]. Despite significant progress, challenges remain in optimizing CAR T therapy for solid tumors, necessitating further research and development to unlock its full potential in cancer treatment.

In one notable study by Feng et al. conducted in 2017, an innovative approach was explored for treating advanced unresectable/metastatic CCA [65]. The research focused on the use of EGFR- and CD133-specific CAR T sequential treatments as a cocktail immunotherapy for CCA patients. This strategy resulted in a clinically significant outcome, with an 8.5-month partial response observed following the EGFR-specific CAR T cell treatment. This was complemented by an additional 4.5-month partial response after the administration of CD133-specific CAR T cells. However, it is essential to thoroughly investigate potential side effects, especially those related to epidermal and endothelial damage caused by CAR T cell infusion, to ensure the therapy’s safety and effectiveness. In a different study, CAR T cells were developed to target EGFR-positive advanced unresectable, relapsed/metastatic CCA. The CART-EGFR cell therapy, administered after conditioning with nab-paclitaxel and cyclophosphamide, demonstrated favorable safety and activity profiles. Patients with EGFR-positive CCA received one to three cycles of CART-EGFR cell infusion, resulting in one complete response and ten cases of stable disease, with a median progression-free survival of 4 months. Notably, the study emphasized the importance of T cell memory subsets, particularly central memory T cells (Tcm), within the infused CART-EGFR cells for improved clinical outcomes. While on-target/off-tumor toxicities were observed, they were manageable, supporting the potential of this therapy. These findings offer a promising approach for treating EGFR-positive advanced CCA, though further research is needed to optimize conditioning regimens and enhance the overall efficacy of CAR T cell therapy in solid tumors [90].

Mao et al. [9] focused on the potential therapeutic target Tn-MUC1 in iCCA. The study found a correlation between Tn-MUC1 expression and poor prognosis in ICC patients and developed effective CAR T cells specifically targeting Tn-MUC1-positive ICC tumors. Despite the absence of a Tn-MUC1-positive ICC cell line, the researchers detected Tn-MUC1 in a significant percentage of ICC patient tumor tissues. Their results demonstrated that CAR T cells engineered with the 5E5 antibody, a monoclonal antibody specifically designed to target the Tn antigen on MUC1—a tumor-associated form of the MUC1 protein often abnormally glycosylated in cancers—effectively eliminated Tn-MUC1-positive ICC (intrahepatic cholangiocarcinoma) cells in both in vitro and in vivo experiments. Moreover, the study suggested that the 5E5 CAR might have broader targeting capabilities beyond Tn-MUC1, potentially making it applicable to various tumor types with different Tn antigens. This research highlights the promise of CAR T cell therapy for ICC and the importance of exploring humanized antibodies with enhanced specificity and affinity for future clinical applications, particularly for solid tumors, where infiltrating the tumor site and overcoming the suppressive tumor microenvironment remain significant challenges [9].

Concurrent upregulation of PD-L1, an immune checkpoint protein, in CCA cells has presented challenges. To address this, CAR T cells were engineered with a PD-1-CD28 switch receptor (αM.CAR/SR T cells), which enhanced cytotoxicity against CCA cells expressing both MUC1 and PD-L1. This innovative approach offers potential for further development in CCA therapy [125]. In a study by Supimon et al. [66], researchers aimed to develop an effective CAR T cell therapy for CCA. They designed a fourth-generation CAR (CAR4) construct targeting the MUC1 antigen, which is overexpressed in CCA and associated with a poor prognosis. CAR4 T cells were engineered to contain three co-stimulatory domains (CD28, CD137, and CD27) linked to CD3ζ to enhance their anti-tumor functions. The study demonstrated that these anti-MUC1-CAR4 T cells led to increased production of pro-inflammatory cytokines (TNF-α and IFN-γ) and granzyme B when exposed to MUC1-expressing CCA cells, leading to specific killing of CCA cells both in 2D and 3D spheroid cultures. These findings suggest the potential of anti-MUC1-CAR4 T cells as a promising immunotherapeutic approach for CCA [66].

CAR T cell therapy targeting integrin αvβ6 in CCA has also shown promise. Integrin αvβ6 was found to be significantly overexpressed in CCA patient tissues, and CAR constructs effectively exhibited cytotoxicity against CCA cells. The study’s results show the potential of integrin αvβ6-specific CAR T cell therapy, with A20-4G CAR T cells demonstrating advantages in mitigating cytokine release syndrome (CRS) [115]. Recent advancements in CAR T cell therapy for CCA have explored a variety of antigen targets, demonstrating promising strides across several preclinical and clinical studies. A study targeting the olfactory receptor OR2H1 with CAR T cells showed specific cytotoxic activity against tumor cells and inhibited tumor growth, providing significant insights into the potential of targeting less conventional antigens [126]. Following this, Sangsuwannukul et al. explored CAR T cells targeting CD133, and demonstrated that these cells can effectively eradicate CD133-expressing CCA cells, significantly increasing tumor cell lysis and enhancing key immune response cytokines IFN-γ and TNF-α [52]. Table 3 provides a summary of both preclinical and clinical research studies focused on various CAR T cell therapies for CCA, illustrating the broad spectrum of ongoing efforts and their implications.

Innovations in CAR T cell design, such as the development of fourth-generation CARs and the exploration of simultaneous targeting of multiple antigen targets, suggest a nuanced approach to improving the efficacy and specificity of these therapies in CCA. Methods for dual antigen targeting include creating a cocktail or sequential infusion of separate CAR T products, co-transduction, and using a bicistronic CAR, a bivalent tandem CAR (targeting two antigens in sequence), or a bivalent loop CAR [127]. Especially in CCA, where tumor antigen expression is heterogeneous, targeting multiple antigens with CAR T cells may lead to improved recognition and efficacy of the therapy and overcoming antigen loss, while maintaining acceptable toxicity levels. Additionally, the deletion of negative regulators of CAR T cell function and immune checkpoints in CAR T cells using gene-editing technologies such as CRISPR/Cas9 can serve as effective strategies in preventing anergy and improving CAR T cell efficacy [128]. For example, PD-1 knockout in mesothelin-targeting CAR T cells was shown to increase cytokine production, improve tumor control, and prevent breast cancer in breast cancer [129], PD-1-disrupted GPC3-CAR T cells have shown enhanced in vivo anti-tumor activity for HCC by targeting GPC3-expressing tumor cells and overcoming immune suppression through PD-1 pathway disruption [130].


cancers-16-03232-t003_Table 3Table 3Summary table of preclinical and clinical research studies focused on chimeric antigen receptor (CAR) T cell therapies for CCA. Abbreviations: chimeric antigen receptor 4th generation (CAR4 T), epidermal growth factor receptor (EGFR), interleukin (IL), intrahepatic cholangiocarcinoma (ICC), mucin 1 (MUC1), natural killer (NK), programmed cell death protein (PD-1), T cell immunoglobulin and ITIM domain (Tigit), T cell immunoglobulin and mucin domain-containing-3 (TIM-3), transforming growth factor-β (TGF-β).

Pre-ClinicalAuthorTitleDateSummaryPhase, Enrollment StatusResultsRef.Sangsuwannuku T et al.Anti-tumor effect of the fourth-generation chimeric antigen receptor T cells targeting CD133 against CCA cells2020Anti-CD133-CAR4 T cells can be used to target CD133-expressing cancers as an alternative cellular immunotherapy in CD133-positive CCA, and may also be beneficial for treating other CD133-expressing cancers.

[52]Supimon K et al.Anti-mucin 1 chimeric antigen receptor T cells for adoptive T cell therapy of CCA2021Anti-MUC1-CAR4 T cells could effectively disrupt KKU-213A spheroids. 

[66]Phanthaphol N et al.Chimeric Antigen Receptor T Cells Targeting Integrin αvβ6 Expressed on CCA Cells2021A20-4G CAR T cells had lower levels of cytokine production, but with higher proliferation rates; represents a promising potential adoptive T cell therapy for integrin αvβ6-positive CCA.

[115]Supimon K et al.Cytotoxic activity of anti-mucin 1 chimeric antigen receptor T cells expressing PD-1-CD28 switch receptor against CCA cells2023The cytotoxic function of aM.CAR/SR T cells was enhanced over the aM.CAR T cells, which have potential to be further tested for CCA treatment.

[125]Mao L et al.Development of Engineered CAR T-cells Targeting Tumor-Associated Glycoforms of MUC1 for the Treatment of Intrahepatic CCA2023Tn-MUC1 may be a potential therapeutic target for ICC, and its expression level was positively correlated with poor prognosis of ICC patients.

[9]Qiao Y et al.Enhancement of CAR T-cell activity against CCA by simultaneous knockdown of six inhibitory membrane proteins2023PTG-T16R-scFV-CAR T cells with knockdown of sextuplet inhibitory molecules (PD-1, Tim-3, Tigit, TGFβR, IL-10R, IL-6R) exhibited strong anti-tumor effect against CCA and long-term efficacy both in vitro and in vivo. 

[131]Chiawpanit C et al.Precision immunotherapy for cholangiocarcinoma: Pioneering the use of human-derived anti-cMET single chain variable fragment in anti-cMET chimeric antigen receptor (CAR) NK cells2024Anti-cMET CAR-NK cells were developed using a human-derived ScFv to target cMET. These engineered NK cells effectively killed cMET-expressing CCA cells, highlighting their potential as a promising therapy for CCA.

[132]

Animal ModelNebbia M et al. (Abstract)B7-h3 targeted CAR T-cell immunotherapy for primary and metastatic/multi-focal intrahepatic CCA2022B7-H3 CAR T cell therapy is effective in eradicating both primary and multi- focal disease and ICC metastases established in NSG mice and in prolonging their survival.

[133]

Clinical Feng K et al.Cocktail treatment with EGFR-specific and CD133-specific chimeric antigen receptor modified T cells in a patient with advanced CCA2017CAR T cocktail immunotherapy may be feasible for the treatment of CCA as well as other solid malignancies; however, the toxicities, especially the epidermal/endothelial damages, require further investigation.Phase IN = 14-day successive infusion of 2.2 × 10^6^/kg total (CART-EGFR)2nd cycle: 2.1 × 10^6^/kg CART-EGFR1 cycle of CART-EGFR + 2 cycles nivolumab1.22 × 10^6^/kg CD133- CART8.5 month PR from CART-EGFR, 4.5 month PR from CART133[65]Feng K et al.Phase I study of chimeric antigen receptor modified T cells in treating HER2-positive advanced biliary tract cancers and pancreatic cancers2018Data from this study demonstrated the safety and feasibility of CART-HER2 immunotherapy and showed encouraging signals of clinical activity.Phase IN = 4 pCCA, 4 iCCA, 1–2 cycles (median dose: 2.45 × 10^6^/kg)1 partial response (4.5 months PFS), 3 stable disease, 4 progressive diseaseMedian PFS: 3.25 months (range, 1.5–5 months)[99]Guo Y et al.Phase I Study of Chimeric Antigen Receptor–Modified T Cells in Patients with EGFR-Positive Advanced Biliary Tract Cancers2018The CART-EGFR cell immunotherapy was a safe and active strategy for EGFR-positive advanced BTCs. The enrichment of Tcm in the infused CART-EGFR cells could predict clinical response.Phase IN = 14 CCA1–3 cycles (median dose, 2.65 × 10^6^/kg; range, 0.8–4.1 × 10^6^/kg) within 6 months1 complete response, 10 stable diseaseMedian PFS: 4 months (range, 2.5–22 months) [90]


## 9. Tumor-Infiltrating Lymphocytes in Cholangiocarcinoma

TILs, particularly T cells, play a key role in the immune surveillance of the cholangiocarcinoma (CCA) tumor microenvironment, often reflecting an active, though frequently suppressed, immune response against the cancer [33]. While a higher density of TILs is associated with improved patient outcomes, CCA employs immune evasion mechanisms. Researchers are exploring ways to enhance TIL recruitment and activation via immune checkpoint inhibitors and personalized TIL therapy. The proportions and distribution of TILs in CCA show distinct patterns compared to hepatocellular carcinoma (HCC) and healthy liver tissue. In CCA, there is a generally lower number of CD8+ T cells and NK cells, but a higher concentration of Tregs and more pronounced immunoinhibitory checkpoints [48,134]. The most common inflammatory cells in CCA are T lymphocytes, predominantly CD8+ T cells, followed by CD4+ T cells, with B lymphocytes being less frequent and NK cells present in modest numbers [22]. The spatial distribution of immune cells in intrahepatic cholangiocarcinoma (iCC) reveals notable patterns: CD8+, CD4+, and CD3+ T cells primarily surround the tumor, while Foxp3+ T cells sometimes infiltrate it. This distinction in cell localization may significantly influence the efficacy of immune responses and treatment outcomes [135]. In extrahepatic CCA, CD8+ and CD4+ T cells are mainly found in peri-tumoral areas, with Foxp3+ T cells observed in intra-tumoral regions [136]. However, the distribution of these cell types can be inconsistent across studies. For instance, B cells, which are less studied, were shown to infiltrate more in the peri-tumoral than in the intra-tumoral area in one study. Overall, the current literature suggests that CD8+, CD4+, and CD3+ T cells are predominantly located in the peri-tumoral area, regardless of CCA subtype [33]. However, the specific locations of Foxp3+ T cells and B cells require further investigation to clarify their patterns in different contexts. This detailed understanding of TILs in CCA will help to elucidate their potential roles in tumor biology and therapeutic targeting.

In the molecular pathogenesis of CCA related to TILs, several pathways and genes play crucial roles. Genes elevated in the Wnt/β-catenin and TGF-signaling pathways are associated with lower numbers of certain CD8+ TILs [47], while pathways like aPKC-I [137], P-Sp1/Snail [137], and Fas/FasL [138] are involved in immunosuppression and tumor cytotoxicity. The B7-H1/PD-1 pathway contributes to immune evasion by promoting apoptosis in CD8+ TILs [139]. Genetic alterations in *KIR* and *HLA* gene loci affect NK cell tumor surveillance, and *KRAS* mutations are linked to low TIL density and tumor immunogenicity [24]. The roles of cytokines, proteins, such as CXCL9, PRKAR1A, and components in the IL-10 and TGF-β pathways are significant in regulating TIL behavior and effectiveness [33], influencing survival outcomes and offering potential targets for therapy.

In advancing TIL therapies, two methodologies are crucial due to their impact on treatment efficacy and outcomes. The first method involves isolating and expanding cytotoxic T lymphocytes (CTLs) from peripheral blood mononuclear cells (PBMCs), using apoptotic tumor cells to specifically enhance their ability to target cancer cells. This approach not only allows for personalized treatments tailored to individual tumor characteristics but also improves the CTLs’ effectiveness against cancer. The second method uses surgically removed tumor lesions for in vitro TIL expansion at a Good Manufacturing Practice (GMP) facility, rapidly increasing the number of active immune cells. This is essential for producing a sufficient volume of TILs capable of overcoming the tumor’s immune evasion tactics for longer survival rates and better patient outcomes [140]. Both approaches exemplify the shift towards adaptive, targeted cancer therapies that leverage the immune system’s natural capacity to fight tumors. The intricate interplay between TILs and the tumor microenvironment in CCA shows their critical role in mediating the immune response against cancer (Appendix A). Despite the challenges posed by the immunosuppressive landscape of CCA, strategies to enhance TIL recruitment, activation, and therapeutic efficacy show potential for personalized cancer treatments. With advancements in understanding the distribution, function, and molecular pathways influencing TIL behavior, there emerges a promising horizon for leveraging these immune cells in precision immunotherapy.

## 10. T Cell Therapies in Cholangiocarcinoma: Beyond CAR T Cell Therapy

Immunotherapies, particularly T cell-based therapies, have introduced new treatment options for CCA. While chimeric antigen receptor (CAR) T cell therapy has gained attention, other forms of T cell therapy, such as T cell receptor (TCR) therapy, adoptive T cell therapy, and tumor-infiltrating lymphocyte (TIL) therapy, offer promising alternatives for targeting CCA. These therapies expand the range of targetable antigens, including intracellular proteins, potentially improving outcomes for patients with this aggressive cancer [141].

TCR-T cell therapy involves modifying a patient’s T cells to express a specific TCR that recognizes tumor antigens presented by major histocompatibility complex (MHC) molecules. Unlike CAR T cells, which only recognize surface antigens, TCR-T cells can target intracellular proteins that are processed and presented on tumor cell surfaces, broadening the range of antigens they can attack. Several antigens relevant to CCA have been identified as potential targets for TCR-T therapy. For example, NY-ESO-1, a cancer/testis antigen, an antigen expressed in cancer cells and normally restricted to testis tissue is expressed in a variety of cancers, including CCA [142]. Clinical trials using TCR-engineered T cells targeting NY-ESO-1 have shown significant tumor regression [142,143], highlighting its potential in CCA. Similarly, MAGE-A3, another cancer/testis antigen, has been studied in TCR-T cells for its therapeutic benefits [144]. Moreover, KRAS mutations, particularly the G12D variant, are common in CCA and represent another target for TCR therapy. TCRs engineered to recognize mutant KRAS epitopes have demonstrated efficacy in preclinical models for solid tumors [144], underscoring the precision of TCR-T therapy in addressing oncogenic drivers in CCA.

Adoptive T cell therapy, which involves the isolation, expansion, and reinfusion of tumor-specific T cells, is another promising approach for CCA. This therapy can involve the use of naturally occurring T cells, such as those harvested from tumor tissue (TILs), or genetically modified T cells, such as TCR-engineered cells [145]. Tumor-infiltrating lymphocyte (TIL) therapy relies on isolating and expanding T cells that have infiltrated the tumor microenvironment. These T cells are expanded ex vivo and reinfused into the patient, often in combination with cytokines like interleukin-2 (IL-2) to boost their activity. In CCA, TIL therapy could be particularly valuable due to the immunosuppressive tumor microenvironment, which limits the effectiveness of the immune response [146]. Early research suggests that expanding tumor-reactive T cells may enhance immune responses against CCA when used alongside immune checkpoint inhibitors that block inhibitory signals like PD-1/PD-L1.

One of the most promising aspects of TCR-T cell therapy is the ability to target neoantigens, which are tumor-specific antigens arising from somatic mutations. Neoantigens are unique to each tumor and represent ideal targets for personalized immunotherapies [147]. Advances in genomic sequencing have made it possible to identify neoantigens in CCA, leading to the development of TCRs that can specifically recognize these antigens. By targeting neoantigens, TCR-T cells can potentially overcome the challenge of tumor heterogeneity, which is a major obstacle in treating CCA. The personalized nature of neoantigen-specific TCRs ensures precise targeting of the patient’s tumor, reducing off-target effects and improving efficacy.

Although the development of TCR-T and TIL therapies for CCA is still in the early stages, the potential of these therapies is undeniable. Ongoing research is focused on identifying new target antigens, optimizing T cell expansion and persistence, and overcoming the immunosuppressive barriers present in the tumor microenvironment. The integration of T cell therapies into clinical practice will likely depend on combination strategies, such as pairing T cell therapies with immune checkpoint inhibitors or cytokine therapies to enhance their function and durability. These approaches aim to improve long-term therapeutic outcomes in CCA by enhancing the body’s immune response to the tumor. T cell therapies, including TCR-T and TIL therapies, represent a promising frontier in the treatment of cholangiocarcinoma. By targeting a broader range of tumor antigens, including intracellular proteins and neoantigens, these therapies complement CAR T cell therapy and offer hope for overcoming some of the key challenges in treating CCA, such as tumor heterogeneity and immune evasion. As research continues to advance, these therapies hold the potential to significantly improve outcomes for patients.

Traditional therapies for CCA are discussed here. In conjunction with the advances of CAR T cell therapy, there have been advances in traditional treatment modalities (chemotherapy, radiation therapy) that should be considered in the multidisciplinary care of patients with CCA. For patients with advanced or metastatic iCCA, treatment with gemcitabine plus cisplatin is first-line, as the ABC-02 phase II trial showed improved overall survival (median OS: 11.7 months versus 8.1 months) compared to gemcitabine alone [148]. For patients with resectable disease and high-risk features, triple regimen neoadjuvant therapy with gemcitabine, cisplatin, and PD-L1 inhibitor durvalumab was shown to have improved progression-free and overall survival in the TOPAZ—1 trial (median OS: 12.8 months versus 11.5 months) when compared to gemcitabine/cisplatin/placebo [149]. Other immune checkpoint inhibitors, such as pembrolizumab, which target PD-1, have shown preliminary success in prolonging overall survival (NCT04003636). Other ICIs are currently being explored in clinical trials, such as toripalimab (anti-PD-1), nivolumab (anti-PD-1), avelumab (anti-PD-L1), tremelimumab (anti-CTLA-4), and ipilumumab (anti-CTLA-4) [58]. Following resection, the BILCAP trial showed prolonged overall survival with the use of adjuvant capecitabine (overall survival: 51.1 months versus 36.4 months) [150].

With the advancement of precision medicine, targeted inhibitors may be successful for patients with known genetic alterations. For example, futibatinib, a tyrosine kinase inhibitor against FGFR, was shown to have an objective response rate of 42% and disease control rate of 83% in 103 patients with previously treated iCCA and known FGFR2 genetic alteration. FGFR inhibitor pemigatinib has also been FDA-approved for previously treated, unresectable, locally advanced, or metastatic CCA with FGFR2 fusions. IDH1 inhibitors like ivosidenib have also shown prolonged median overall survival (10.3 months versus 7.5 months) compared to placebo for patients with known IDH1 mutations [151]. Knowledge of targetable mutations may allow for improved selection of agents for combination with immune-based therapy.

## 11. Future Prospects: T Cell Therapy Revolutionizing CCA Treatment

Innovations in genomic studies have illuminated the genetic mutations and pathways that drive CCA, facilitating the development of personalized therapies, including checkpoint inhibitors and T cell therapies tailored to specific molecular profiles. Clinical trials are pivotal in evaluating these innovative treatments, with patient participation helping to refine their efficacy and safety. The emphasis on patient-centric approaches ensures that research and trials align with patient needs, enhancing treatment outcomes. International collaboration, supported by robust funding from various sectors, and open science initiatives accelerate progress by promoting data sharing and transparency. This collective effort across stakeholders—including researchers, clinicians, patients, and funders—is crucial for advancing treatment paradigms, leading to improved detection, targeted therapies, and better outcomes for CCA patients, marking a hopeful step forward in the battle against this challenging malignancy.

## 12. Conclusions

T cell therapies, including TCR-T and TIL therapies, offer promising advancements in the treatment of cholangiocarcinoma by expanding the range of antigens that can be targeted. While CAR T cell therapy has primarily focused on surface antigens, TCR-T therapy introduces the ability to target intracellular proteins and tumor-specific mutations, addressing some of the limitations posed by tumor heterogeneity in CCA. In parallel, TIL therapy utilizes the body’s own immune cells to enhance the natural immune response against tumors, offering an alternative approach for tumors with an immunosuppressive microenvironment.

Though still in the early stages for cholangiocarcinoma, these therapies provide additional options that may complement existing treatments. By focusing on precision and expanding the range of targetable antigens, T cell therapies are emerging as a valuable tool in the evolving landscape of CCA treatment. Continued research and clinical trials will be key to determining their long-term efficacy and potential for integration into current therapeutic strategies.

## Figures and Tables

**Figure 1 cancers-16-03232-f001:**
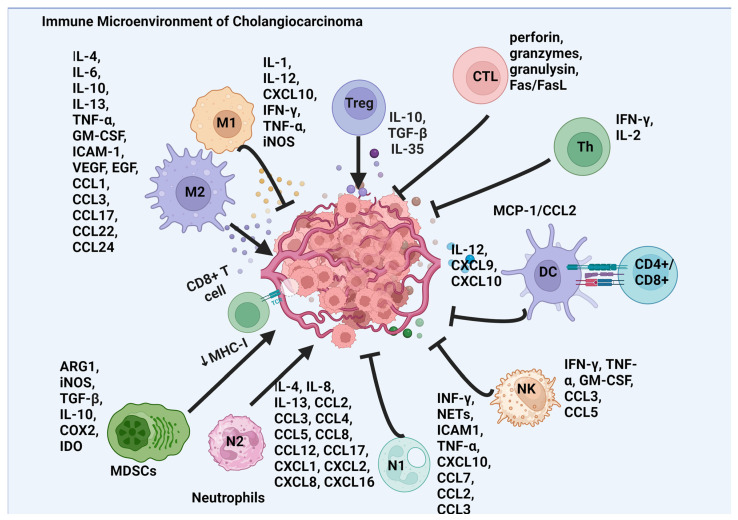
Schematic of the intricate immune microenvironment in CCA, mapping the interplay between different immune cells and their associated mediators. The diagram displays the tumor in the center, surrounded by various immune cells: M1 and M2 macrophages, which release pro-inflammatory and anti-inflammatory cytokines, respectively; T-regulatory cells (Tregs), which produce immunosuppressive cytokines; cytotoxic T lymphocytes (CTLs), which secrete cytotoxic molecules; helper T cells (Th), which aid in immune response modulation; dendritic cells (DCs), which present antigens; natural killer (NK) cells, which release cytotoxic substances; and myeloid-derived suppressor cells (MDSCs), which contribute to tumor growth and immune evasion. Additionally, the graphic outlines the complex network of cytokines and chemokines, such as interleukins, tumor necrosis factors, and chemotactic cytokines (CCLs and CXCLs), highlighting the multifaceted interactions that contribute to the cancer’s immune environment. Abbreviations: arginase 1 (ARG1), chemokine (C-C motif) ligand (CCL), CXC motif ligand (CXCL), cyclo-oxygenase (COX), epidermal growth factor (EGF), interferon (IFN), indoleamine 2,3-dioxygenase (IDO), intracellular adhesion molecule-1 (ICAM-1), granulocyte–macrophage colony-stimulating factor (GM-CSF), major histocompatibility complex (MHC), neutrophil extracellular traps (NETs), tumor necrosis factor (TNF), transforming growth factor-β (TGF-β), vascular endothelial growth factor (VEGF).

**Figure 2 cancers-16-03232-f002:**
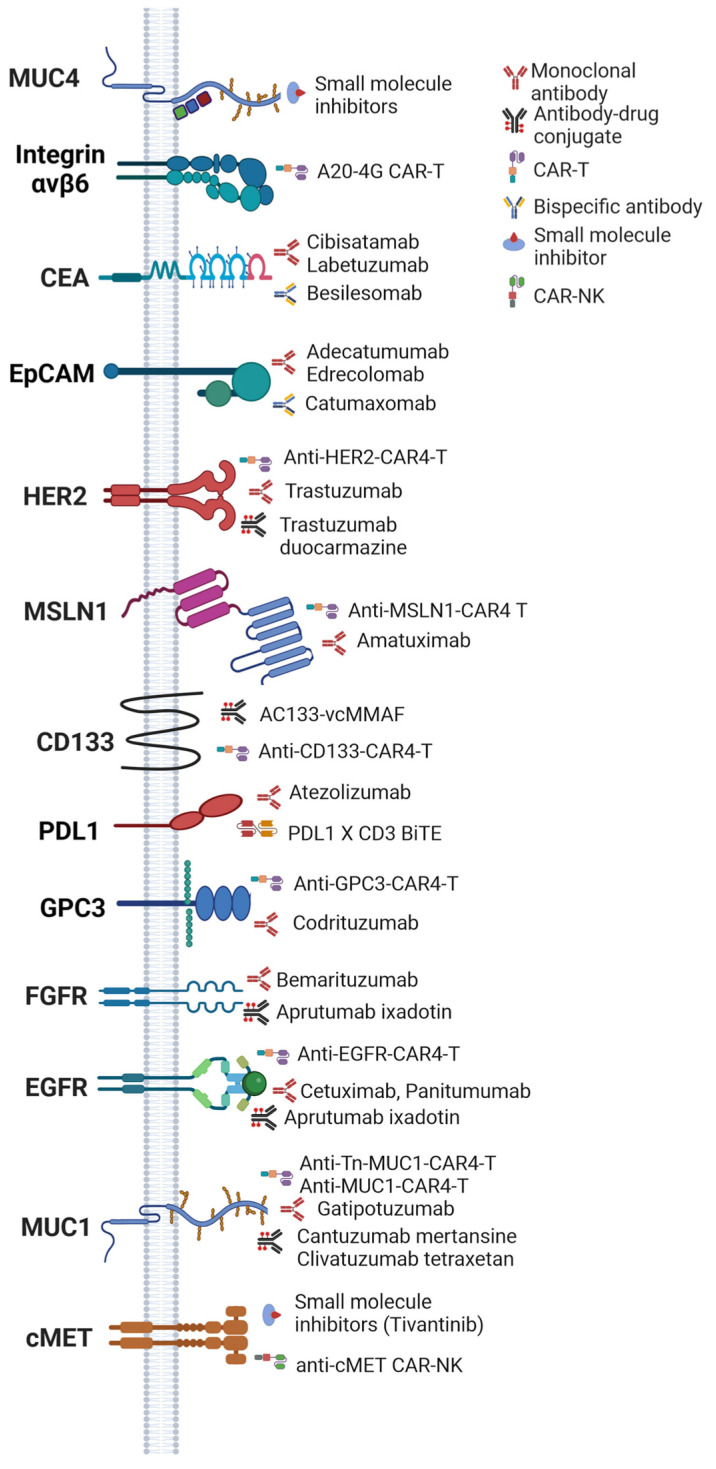
Image showing the different surface antigens and their corresponding targeting immunotherapies used in different clinical trials and/or preclinical models of CCA. Created with BioRender.com. Abbreviations: carcinoembryonic antigen (CEA), chimeric antigen receptor therapy (CAR T), chimeric antigen receptor natural killer (CAR-NK), epithelial cell adhesion molecule (EpCAM), epidermal growth factor receptor (EGFR), fibroblast growth factor receptor (FGFR), glypican-3 (GPC3), human epidermal growth factor 2 (HER2), mucin 1 (MUC1), mucin 4 (MUC4), mesothelin 1 (MSLN1), programmed cell death ligand 1 (PD-L1).

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
