# Peer review of "Advancing Cholangiocarcinoma Care: Insights and Innovations in T Cell Therapy"

_cancers, 2024, doi:10.3390/cancers16183232_

Round 1

Reviewer 1 Report

Comments and Suggestions for Authors

The manuscript is a review article regarding to T cell therapy in cholangiocarcinoma(CCA).

The authors described several promising results in T cells and CCA. The conclusion in abstract and text seemed not conclusive.

The are some comments as following:

1.the descriptions of CCA seemed to be well and in details.

2.the abstract would be totally revised and clearly concluded the present progress and clinical results of T cell therapy.

3.could the authors provide references in table 1.

4.the most important part of the manuscript is table 3. The human studies should be extended with some items, such as phase in the trials, enrolled patient numbers, major study areas, survival or treatment response parameters.

Comments on the Quality of English Language

minor

Reviewer 2 Report

Comments and Suggestions for Authors

cancers-3194587

Type of manuscript: Review

Title: Advancing Cholangiocarcinoma Care: Insights and Innovations in T Cell Therapy

Authors: Neda Dadgar, Arun K Arunachalam, Hanna Hong, Yee Peng Phoon, Jorge E Arpi-Palacios, Melis Uysal, Chase J Wehrle, Federico Aucejo, Wen Wee Ma, Jan Joseph Melenhorst *

This paper is a review on the potential of CAR T-cell therapy as a treatment for advanced cholangiocarcinoma (CCA) and is generally well-written. However, due to the extensive coverage of the topic, there are some instances of verbosity in the logical flow, and there are errors in the consistent use of certain terms throughout the manuscript.

[Primary concerns]

1.   CCA is the second most common primary liver cancer after hepatocellular carcinoma, accounting for approximately 15% as the authors noted. While the authors only mentioned the incidence in the paper, it is also important to address the high mortality rate due to CCA, especially in East Asia, where the death rate from CCA is notably high globally.

2.   In addition to CAR T-cell therapy, T cell therapy also includes T-cell Receptor (TCR) therapy. Several studies have been published on TCR therapy in relation to CCA, and the content of these studies should be added.

3.   In addition to T cell therapy for the treatment of CCA, combination therapies utilizing traditional approaches such as chemotherapy, radiotherapy, and other immunotherapies are also viable options. Content related to these combination strategies should be included.

4.   It is also necessary to include content related to the latest technologies associated with CCA treatment, particularly CRISPR-Cas9-based gene editing to enhance T cell function and new antigen targeting approaches.

5.   Abbreviations: The use of abbreviations when writing a review paper has many advantages besides simplicity of expression. To use an abbreviation, first write the abbreviation in parentheses after the full name, and then use the abbreviation from Introduction to the final Conclusion. Only in Abstract and Figure legend do it separately. If an abbreviation is not used more than twice, there is no need to define it, so please delete it. Many of the abbreviations used in the paper have been repeatedly defined, even though they were initially defined earlier. Please correct all such instances.

6.   In cases where abbreviations are used within figures or tables, please list these abbreviations along with their corresponding full names in the figure legends or at the bottom of corresponding tables. If there are two or more abbreviations, arrange them in alphabetical order. When listing the full names of abbreviations, do not capitalize the first letter of each word unless they are proper nouns.

[Minor concerns]

1.   Line 73: ‘CCA tumors’ should be written as ‘CCA’. CCA itself is a malignant tumor.

2.   Line 136: Since cholangiocarcinoma had already been abbreviated at Line 44 as CCA, just write CCA here.

3.   Line 147: Since ‘tumor microenvironment’ had already been abbreviated at Line 85 as TME, just write TME here. The same mistake is also found in line 156.

4.   Line 166: IL6 and IL10 should be written as IL-6 and IL-10.

5.   Figure 1: Re-write the following key terms properly: TNF-a, IFN-g, IFNg, TGFb, etc.

6.   Line 263: CCA is enough in the sentence.

7.   Line 309: ‘Adoptive T cell Therapy’ should be written as ‘adoptive T cell therapy’. Do not use unnecessary capitalization. Similar typos are also found in lines 369 and 371; please correct them as well.

8.   Line 337: The term 'bile duct cancer (CCA)' is used, but is it necessary to phrase it this way? Either use just 'CCA' or consider a different phrasing.

9.   Table 2: The antigen names are listed with a mix of using abbreviations first followed by the full name in parentheses, and vice versa. Please change all entries to list the full name first, followed by the abbreviation in parentheses.

10.    Figure 2: CAR NK vs. CAR-NK; CAR T vs. CAR-T. Use a consistent notation throughout.

11.    Lines 436 and 444: iCCA is enough. It had already been abbreviated at Line 95.

12.    Line 460: ‘toenhance’ should be ‘to enhance’.

13.    Line 475: When citing references using author names within the text, include only the last name of the first author.

14.    Line 519: Re-write TGF-b properly.

15.    Some of the references are missing volume numbers or page numbers, so please correct them. Examples: 27, 29, 43, 47, 54, 64, 69, 100, 107, 126, etc. Double-check the format, too.

Overall, the manuscript can be considered to publication after major revision as indicated above.

Comments on the Quality of English Language

cancers-3194587

Type of manuscript: Review

Title: Advancing Cholangiocarcinoma Care: Insights and Innovations in T Cell Therapy

Authors: Neda Dadgar, Arun K Arunachalam, Hanna Hong, Yee Peng Phoon, Jorge E Arpi-Palacios, Melis Uysal, Chase J Wehrle, Federico Aucejo, Wen Wee Ma, Jan Joseph Melenhorst *

This paper is a review on the potential of CAR T-cell therapy as a treatment for advanced cholangiocarcinoma (CCA) and is generally well-written. However, due to the extensive coverage of the topic, there are some instances of verbosity in the logical flow, and there are errors in the consistent use of certain terms throughout the manuscript.

[Primary concerns]

1.   CCA is the second most common primary liver cancer after hepatocellular carcinoma, accounting for approximately 15% as the authors noted. While the authors only mentioned the incidence in the paper, it is also important to address the high mortality rate due to CCA, especially in East Asia, where the death rate from CCA is notably high globally.

2.   In addition to CAR T-cell therapy, T cell therapy also includes T-cell Receptor (TCR) therapy. Several studies have been published on TCR therapy in relation to CCA, and the content of these studies should be added.

3.   In addition to T cell therapy for the treatment of CCA, combination therapies utilizing traditional approaches such as chemotherapy, radiotherapy, and other immunotherapies are also viable options. Content related to these combination strategies should be included.

4.   It is also necessary to include content related to the latest technologies associated with CCA treatment, particularly CRISPR-Cas9-based gene editing to enhance T cell function and new antigen targeting approaches.

5.   Abbreviations: The use of abbreviations when writing a review paper has many advantages besides simplicity of expression. To use an abbreviation, first write the abbreviation in parentheses after the full name, and then use the abbreviation from Introduction to the final Conclusion. Only in Abstract and Figure legend do it separately. If an abbreviation is not used more than twice, there is no need to define it, so please delete it. Many of the abbreviations used in the paper have been repeatedly defined, even though they were initially defined earlier. Please correct all such instances.

6.   In cases where abbreviations are used within figures or tables, please list these abbreviations along with their corresponding full names in the figure legends or at the bottom of corresponding tables. If there are two or more abbreviations, arrange them in alphabetical order. When listing the full names of abbreviations, do not capitalize the first letter of each word unless they are proper nouns.

[Minor concerns]

1.   Line 73: ‘CCA tumors’ should be written as ‘CCA’. CCA itself is a malignant tumor.

2.   Line 136: Since cholangiocarcinoma had already been abbreviated at Line 44 as CCA, just write CCA here.

3.   Line 147: Since ‘tumor microenvironment’ had already been abbreviated at Line 85 as TME, just write TME here. The same mistake is also found in line 156.

4.   Line 166: IL6 and IL10 should be written as IL-6 and IL-10.

5.   Figure 1: Re-write the following key terms properly: TNF-a, IFN-g, IFNg, TGFb, etc.

6.   Line 263: CCA is enough in the sentence.

7.   Line 309: ‘Adoptive T cell Therapy’ should be written as ‘adoptive T cell therapy’. Do not use unnecessary capitalization. Similar typos are also found in lines 369 and 371; please correct them as well.

8.   Line 337: The term 'bile duct cancer (CCA)' is used, but is it necessary to phrase it this way? Either use just 'CCA' or consider a different phrasing.

9.   Table 2: The antigen names are listed with a mix of using abbreviations first followed by the full name in parentheses, and vice versa. Please change all entries to list the full name first, followed by the abbreviation in parentheses.

10.    Figure 2: CAR NK vs. CAR-NK; CAR T vs. CAR-T. Use a consistent notation throughout.

11.    Lines 436 and 444: iCCA is enough. It had already been abbreviated at Line 95.

12.    Line 460: ‘toenhance’ should be ‘to enhance’.

13.    Line 475: When citing references using author names within the text, include only the last name of the first author.

14.    Line 519: Re-write TGF-b properly.

15.    Some of the references are missing volume numbers or page numbers, so please correct them. Examples: 27, 29, 43, 47, 54, 64, 69, 100, 107, 126, etc. Double-check the format, too.

Overall, the manuscript can be considered to publication after major revision as indicated above.

Reviewer 3 Report

Comments and Suggestions for Authors

Here, the authors explore the potential of CAR T-cell therapy for treating cholangiocarcinoma (CCA), a highly aggressive bile duct cancer. The review article underscores the critical role of the tumor microenvironment, which includes the surrounding cells, blood vessels, and molecules that support tumor growth, in the progression of CCA. The article also highlights the significant challenges faced in applying CAR T-cell therapy to solid tumors. It provides a comprehensive summary of the latest research, ongoing clinical trials, and the rapidly evolving landscape of precision medicine. The necessity for continued advancements and innovations in this field is emphasized to improve patient outcomes and offer new hope for those affected by CCA.

Main points

The article highlights the groundbreaking potential of CAR T-cell therapy, offering a new and hopeful treatment avenue for patients with CCA. It provides an in-depth overview of current research, detailing various clinical trials and future directions in the field.

The authors emphasize the importance of personalized treatment approaches, tailored to the unique genetic and molecular profiles of individual patients, which is a cornerstone of precision medicine. 

The authors also discuss the significant difficulties in applying CAR T-cell therapy to solid tumors, including the complex nature of the tumor microenvironment and the need for more effective delivery methods. 

The review acknowledges the current limitations of available therapies for CCA and the generally poor prognosis associated with conventional treatments, highlighting the urgent need for more effective solutions.

Minor points:

For the sake of clarity, I would update Figure 1 with before/ after therapies, and what biomarkers are usually changed with the CAR-T therapy/ immune checkpoint blockades. The biomarkers could also be added to Table 1 as an alternative.

Table 1 should be fully referenced. 

Comments on the Quality of English Language

minor typos

Round 2

Reviewer 1 Report

Comments and Suggestions for Authors

the authors have reivsed the manuscripts according to comments

Comments on the Quality of English Language

minor

Reviewer 2 Report

Comments and Suggestions for Authors

The issues that were pointed out have all been appropriately corrected, so I recommend accepting this review paper.